# Long short-term memory and learning-to-learn in networks of spiking neurons

**Guillaume Bellec\*, Darjan Salaj\*, Anand Subramoney\*, Robert Legenstein & Wolfgang Maass**
Institute for Theoretical Computer Science
Graz University of Technology, Austria
{bellec,salaj,subramoney,legenstein,maass}@igi.tugraz.at

## Abstract

Recurrent networks of spiking neurons (RSNNs) underlie the astounding computing and learning capabilities of the brain. But computing and learning capabilities of RSNN models have remained poor, at least in comparison with artificial neural networks (ANNs). We address two possible reasons for that. One is that RSNNs in the brain are not randomly connected or designed according to simple rules, and they do not start learning as a tabula rasa network. Rather, RSNNs in the brain were optimized for their tasks through evolution, development, and prior experience. Details of these optimization processes are largely unknown. But their functional contribution can be approximated through powerful optimization methods, such as backpropagation through time (BPTT).

A second major mismatch between RSNNs in the brain and models is that the latter only show a small fraction of the dynamics of neurons and synapses in the brain. We include neurons in our RSNN model that reproduce one prominent dynamical process of biological neurons that takes place at the behaviourally relevant time scale of seconds: neuronal adaptation. We denote these networks as LSNNs because of their Long short-term memory. The inclusion of adapting neurons drastically increases the computing and learning capability of RSNNs if they are trained and configured by deep learning (BPTT combined with a rewiring algorithm that optimizes the network architecture). In fact, the computational performance of these RSNNs approaches for the first time that of LSTM networks. In addition RSNNs with adapting neurons can acquire abstract knowledge from prior learning in a Learning-to-Learn (L2L) scheme, and transfer that knowledge in order to learn new but related tasks from very few examples. We demonstrate this for supervised learning and reinforcement learning.

## 1 Introduction

Recurrent networks of spiking neurons (RSNNs) are frequently studied as models for networks of neurons in the brain. In principle, they should be especially well-suited for computations in the temporal domain, such as speech processing, as their computations are carried out via spikes, i.e., events in time and space. But the performance of RSNN models has remained suboptimal also for temporal processing tasks. One difference between RSNNs in the brain and RSNN models is that RSNNs in the brain have been optimized for their function through long evolutionary processes, complemented by a sophisticated learning curriculum during development. Since most details of these biological processes are currently still unknown, we asked whether deep learning is able to mimic these complex optimization processes on a functional level for RSNN models. We used BPTT as the deep learning method for network optimization. Backpropagation has been adapted previously for feed forward networks with binary activations in [1, 2], and we adapted BPTT to work

in a similar manner for RSNNs. In order to also optimize the connectivity of RSNNs, we augmented BPTT with DEEP R, a biologically inspired heuristic for synaptic rewiring [3, 4]. Compared to LSTM networks, RSNNs tend to have inferior short-term memory capabilities. Since neurons in the brain are equipped with a host of dynamics processes on time scales larger than a few dozen ms [5], we enriched the inherent dynamics of neurons in our model by a standard neural adaptation process.

We first show (section 4) that this approach produces new computational performance levels of RSNNs for two common benchmark tasks: Sequential MNIST and TIMIT (a speech processing task). We then show that it makes L2L applicable to RSNNs (section 5), similarly as for LSTM networks. In particular, we show that meta-RL [6, 7] produces new motor control capabilities of RSNNs (section 6). This result links a recent abstract model for reward-based learning in the brain [8] to spiking activity. In addition, we show that RSNNs with sparse connectivity and sparse firing activity of 10-20 Hz (see Fig. 1D, 2D, S1C) can solve these and other tasks. Hence these RSNNs compute with spikes, rather than firing rates.

The superior computing and learning capabilities of LSNNs suggest that they are also of interest for implementation in spike-based neuromorphic chips such as Brainscales [9], SpiNNaker [10], True North [2], chips from ETH Zürich [11], and Loihi [12]. In particular, nonlocal learning rules such as backprop are challenges for some of these neuromorphic devices (and for many brain models). Hence alternative methods for RSNN learning of nonlinear functions are needed. We show in sections 5 and 6 that L2L can be used to generate RSNNs that learn very efficiently even in the absence of synaptic plasticity.

**Relation to prior work:** We refer to [13, 14, 15, 16] for summaries of preceding results on computational capabilities of RSNNs. The focus there was typically on the generation of dynamic patterns. Such tasks are not addressed in this article, but it will be shown in [17] that LSNNs provide an alternative model to [16] for the generation of complex temporal patterns. Huh et al. [15] applied gradient descent to recurrent networks of spiking neurons. There, neurons without a leak were used. Hence, the voltage of a neuron could used in that approach to store information over an unlimited length of time.

We are not aware of previous attempts to bring the performance of RSNNs for time series classification into the performance range of LSTM networks. We are also not aware of any previous literature on applications of L2L to SNNs.

## 2 LSNN model

Neurons and synapses in common RSNN models are missing many of the dynamic processes found in their biological counterparts, especially those on larger time scales. We integrate one of them into our RSNN model: neuronal adaptation. It is well known that a substantial fraction of excitatory neurons in the brain are adapting, with diverse time constants, see e.g. the Allen Brain Atlas for data from the neocortex of mouse and humans. We refer to the resulting type of RSNNs as Long short-term memory Spiking Neural Networks (LSNNs). LSNNs consist of a population $R$ of integrate-and-fire (LIF) neurons (excitatory and inhibitory), and a second population $A$ of LIF excitatory neurons whose excitability is temporarily reduced through preceding firing activity, i.e., these neurons are adapting (see Fig. 1C and Suppl.). Both populations $R$ and $A$ receive spike trains from a population $X$ of external input neurons. Results of computations are read out by a population $Y$ of external linear readout neurons, see Fig. 1C.

Common ways for fitting models for adapting neurons to data are described in [18, 19, 20, 21]. We are using here the arguably simplest model: We assume that the firing threshold $B_j(t)$ of neuron $j$ increases by some fixed amount $\beta/\tau_{a,j}$ for each spike of this neuron $j$, and then decays exponentially back to a baseline value $b_j^0$ with a time constant $\tau_{a,j}$. Thus the threshold dynamics for a discrete time step of $\delta t = 1$ ms reads as follows

$$B_j(t) = b_j^0 + \beta b_j(t), \tag{1}$$

$$b_j(t + \delta t) = \rho_j b_j(t) + (1 - \rho_j) z_j(t), \tag{2}$$

where $\rho_j = \exp(-\frac{\delta t}{\tau_{a,j}})$ and $z_j(t)$ is the spike train of neuron $j$ assuming values in $\{0, \frac{1}{\delta t}\}$. Note that this dynamics of thresholds of adaptive spiking neurons is similar to the dynamics of the state of context neurons in [22]. It generally suffices to place the time constant of adapting neurons into the desired range for short-term memory (see Suppl. for specific values used in each experiment).

# 3 Applying BPTT with DEEP R to RSNNs and LSNNs

We optimize the synaptic weights, and in some cases also the connectivity matrix of an LSNN for specific ranges of tasks. The optimization algorithm that we use, backpropagation through time (BPTT), is not claimed to be biologically realistic. But like evolutionary and developmental processes, BPTT can optimize LSNNs for specific task ranges. Backpropagation (BP) had already been applied in [1] and [2] to feedforward networks of spiking neurons. In these approaches, the gradient is backpropagated through spikes by replacing the non-existent derivative of the membrane potential at the time of a spike by a pseudo-derivative that smoothly increases from 0 to 1, and then decays back to 0. We reduced ("dampened") the amplitude of the pseudo-derivative by a factor $< 1$ (see Suppl. for details). This enhances the performance of BPTT for RSNNs that compute during larger time spans, that require backpropagation through several 1000 layers of an unrolled feedforward network of spiking neurons. A similar implementation of BPTT for RSNNs was proposed in [15]. It is not yet clear which of these two versions of BPTT work best for a given task and a given network.

In order to optimize not only the synaptic weights of a RSNN but also its connectivity matrix, we integrated BPTT with the biologically inspired [3] rewiring method DEEP R [4] (see Suppl. for details). DEEP R converges theoretically to an optimal network configuration by continuously updating the set of active connections [23, 3, 4].

# 4 Computational performance of LSNNs

**Sequential MNIST:** We tested the performance of LSNNs on a standard benchmark task that requires continuous updates of short term memory over a long time span: sequential MNIST [24, 25]. We compare the performance of LSNNs with that of LSTM networks. The size of the LSNN, in the case of full connectivity, was chosen to match the number of parameters of the LSTM network. This led to 120 regular spiking and 100 adaptive neurons (with adaptation time constant $\tau_a$ of 700 ms) in comparison to 128 LSTM units. Actually it turned out that the sparsely connected LSNN shown in Fig. 1C, which was generated by including DEEP R in BPTT, had only 12% of the synaptic connections but performed better than the fully connected LSNN (see "DEEP R LSNN" versus "LSNN" in Fig. 1B).

The task is to classify the handwritten digits of the MNIST dataset when the pixels of each handwritten digit are presented sequentially, one after the other in 784 steps, see Fig. 1A. After each presentation of a handwritten digit, the network is required to output the corresponding class. The grey values of pixels were given directly to artificial neural networks (ANNs), and encoded by spikes for RSNNs. We considered both the case of step size 1 ms (requiring 784 ms for presenting the input image) and 2 ms (requiring 1568 ms for each image, the adaptation time constant $\tau_a$ was set to 1400 ms in this case, see Fig. 1B.). The top row of Fig. 1D shows a version where the grey value of the currently presented pixel is encoded by population coding through the firing probability of the 80 input neurons. Somewhat better performance was achieved when each of the 80 input neurons is associated with a particular threshold for the grey value, and this input neuron fires whenever the grey value crosses its threshold in the transition from the previous to the current pixel (this input convention is chosen for the SNN results of Fig. 1B). In either case, an additional input neuron becomes active when the presentation of the 784 pixel values is finished, in order to prompt an output from the network. The firing of this additional input neuron is shown at the top right of the top panel of Fig. 1D. The softmax of 10 linear output neurons $Y$ is trained through BPTT to produce, during this time segment, the label of the sequentially presented handwritten digit. We refer to the yellow shading around 800 ms of the output neuron for label 3 in the plot of the dynamics of the output neurons $Y$ in Fig. 1D. This output was correct.

A performance comparison is given in Fig. 1B. LSNNs achieve 94.7% and 96.4% classification accuracy on the test set when every pixel is presented for 1 and 2ms respectively. An LSTM network achieves 98.5% and 98.0% accuracy on the same task setups. The LIF and RNN bars in Fig. 1B show that this accuracy is out of reach for BPTT applied to spiking or nonspiking neural networks without enhanced short term memory capabilities. We observe that in the sparse architecture discovered by DEEP R, the connectivity onto the readout neurons Y is denser than in the rest of the network (see Fig. 1C). Detailed results are given in the supplement.

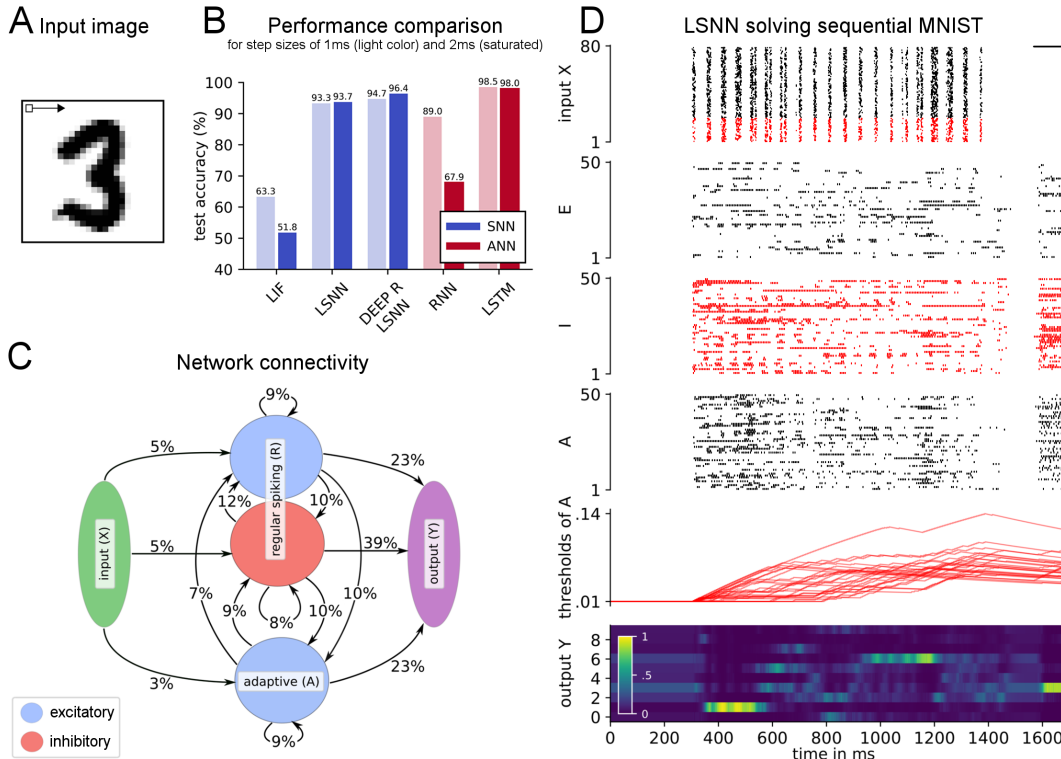

Figure 1: **Sequential MNIST. A** The task is to classify images of handwritten digits when the pixels are shown sequentially pixel by pixel, in a fixed order row by row. **B** The performance of RSNNs is tested for three different setups: without adapting neurons (LIF), a fully connected LSNN, and an LSNN with randomly initialized connectivity that was rewired during training (DEEP R LSNN). For comparison, the performance of two ANNs, a fully connected RNN and an LSTM network are also shown. **C** Connectivity (in terms of connection probabilities between and within the 3 subpopulations) of the LSNN after applying DEEP R in conjunction with BPTT. The input population $X$ consisted of 60 excitatory and 20 inhibitory neurons. Percentages on the arrows from $X$ indicate the average connection probabilities from excitatory and inhibitory neurons. **D** Dynamics of the LSNN after training when the input image from A was sequentially presented. From top to bottom: spike rasters from input neurons (X), and random subsets of excitatory (E) and inhibitory (I) regularly spiking neurons, and adaptive neurons (A), dynamics of the firing thresholds of a random sample of adaptive neurons; activation of softmax readout neurons.

**Speech recognition (TIMIT):** We also tested the performance of LSNNs for a real-world speech recognition task, the TIMIT dataset. A thorough study of the performance of many variations of LSTM networks on TIMIT has recently been carried out in [26]. We used exactly the same setup which was used there (framewise classification) in order to facilitate comparison. We found that a standard LSNN consisting of 300 regularly firing (200 excitatory and 100 inhibitory) and 100 excitatory adapting neurons with an adaptation time constant of 200 ms, and with $20\%$ connection probability in the network, achieved a classification error of $33.2\%$. This error is below the mean error around $40\%$ from 200 trials with different hyperparameters for the best performing (and most complex) version of LSTMs according to Fig. 3 of [26], but above the mean of $29.7\%$ of the 20 best performing choices of hyperparameters for these LSTMs. The performance of the LSNN was however somewhat better than the error rates achieved in [26] for a less complex version of LSTMs without forget gates (mean of the best 20 trials: $34.2\%$).

We could not perform a similarly rigorous search over LSNN architectures and meta-parameters as was carried out in [26] for LSTMs. But if all adapting neurons are replaced by regularly firing excitatory neurons one gets a substantially higher error rate than the LSNN with adapting neurons: $37\%$. Details are given in the supplement.

# 5 LSNNs learn-to-learn from a teacher

One likely reason why learning capabilities of RSNN models have remained rather poor is that one usually requires a tabula rasa RSNN model to learn. In contrast, RSNNs in the brain have been optimized through a host of preceding processes, from evolution to prior learning of related tasks, for their learning performance. We emulate a similar training paradigm for RSNNs using the L2L setup. We explore here only the application of L2L to LSNNs, but L2L can also be applied to RSNNs without adapting neurons [27]. An application of L2L to LSNNs is tempting, since L2L is most commonly applied in machine learning to their ANN counterparts: LSTM networks see e.g. [6, 7]. LSTM networks are especially suited for L2L since they can accommodate two levels of learning and representation of learned insight: Synaptic connections and weights can encode, on a higher level, a learning algorithm and prior knowledge on a large time-scale. The short-term memory of an LSTM network can accumulate, on a lower level of learning, knowledge during the current learning task. It has recently been argued [8] that the pre-frontal cortex (PFC) similarly accumulates knowledge during fast reward-based learning in its short-term memory, without using dopamine-gated synaptic plasticity, see the text to Suppl. Fig. 3 in [8]. The experimental results of [28] suggest also a prominent role of short-term memory for fast learning in the motor cortex.

The standard setup of L2L involves a large, in fact in general infinitely large, family $\mathcal{F}$ of learning tasks $C$. Learning is carried out simultaneously in two loops (see Fig. 2A). The *inner loop* learning involves the learning of a single task $C$ by a neural network $\mathcal{N}$, in our case by an LSNN. Some parameters of $\mathcal{N}$ (termed hyper-parameters) are optimized in an *outer loop* optimization to support fast learning of a randomly drawn task $C$ from $\mathcal{F}$. The outer loop training – implemented here through BPTT – proceeds on a much larger time scale than the inner loop, integrating performance evaluations from many different tasks $C$ of the family $\mathcal{F}$. One can interpret this outer loop as a process that mimics the impact of evolutionary and developmental optimization processes, as well as prior learning, on the learning capability of brain networks. We use the terms training and optimization interchangeably, but the term training is less descriptive of the longer-term evolutionary processes we mimic. Like in [29, 6, 7] we let all synaptic weights of $\mathcal{N}$ belong to the set of hyper-parameters that are optimized through the outer loop. Hence the network is forced to encode all results from learning the current task $C$ in its internal state, in particular in its firing activity and the thresholds of adapting neurons. Thus the synaptic weights of the neural network $\mathcal{N}$ are free to encode an efficient *algorithm* for learning arbitrary tasks $C$ from $\mathcal{F}$.

When the brain learns to predict sensory inputs, or state changes that result from an action, this can be formalized as learning from a teacher (i.e., supervised learning). The teacher is in this case the environment, which provides – often with some delay – the target output of a network. The L2L results of [29] show that LSTM networks can learn nonlinear functions from a teacher without modifying their synaptic weights, using their short-term memory instead. We asked whether this form of learning can also be attained by LSNNs.

**Task:** We considered the task of learning complex non-linear functions from a teacher. Specifically, we chose as family $\mathcal{F}$ of tasks a class of continuous functions of two real-valued variables $(x_1, x_2)$. This class was defined as the family of all functions that can be computed by a 2-layer artificial neural network of sigmoidal neurons with 10 neurons in the hidden layer, and weights and biases from [-1, 1], see Fig. 2B. Thus overall, each such target network (TN) from $\mathcal{F}$ was defined through 40 parameters in the range [-1, 1]: 30 weights and 10 biases. We gave the teacher input to the LSNN for learning a particular TN $C$ from $\mathcal{F}$ in a delayed manner as in [29]: The target output value was given after $\mathcal{N}$ had provided its guessed output value for the preceding input.

This delay of the feedback is consistent with biologically plausible scenarios. Simultaneously, having a delay for the feedback prevents $\mathcal{N}$ from passing on the teacher value as output without first producing a prediction on its own.

**Implementation:** We considered a LSNN $\mathcal{N}$ consisting of 180 regularly firing neurons (population R) and 120 adapting neurons (population A) with a spread of adaptation time constants sampled uniformly between 1 and 1000 ms and with full connectivity. Sparse connectivity in conjunction with rewiring did not improve performance in this case. All neurons in the LSNN received input from a population $X$ of 300 external input neurons. A linear readout received inputs from all neurons in R and A. The LSNN received a stream of 3 types of external inputs (see top row of Fig. 2D): the values of $x_1, x_2$, and of the output $C(x'_1, x'_2)$ of the TN for the preceding input pair $x'_1, x'_2$ (set to 0

at the first trial), all represented through population coding in an external population of 100 spiking neurons. It produced outputs in the form of weighted spike counts during 20 ms windows from all neurons in the network (see bottom row of Fig. 2D), where the weights for this linear readout were trained, like all weights inside the LSNN, in the outer loop, and remained fixed during learning of a particular TN.

The training procedure in the outer loop of L2L was as follows: Network training was divided into training episodes. At the start of each training episode, a new target network TN was randomly chosen and used to generate target values $C(x_1, x_2) \in [0, 1]$ for randomly chosen input pairs $(x_1, x_2)$. 500 of these input pairs and targets were used as training data, and presented one per step to the LSNN during the episode, where each step lasted 20 ms. LSNN parameters were updated using BPTT to minimize the mean squared error between the LSNN output and the target in the training set, using gradients computed over batches of 10 such episodes, which formed one iteration of the outer loop. In other words, each weight update included gradients calculated on the input/target pairs from 10 different TNs. This training procedure forced the LSNN to adapt its parameters in a way that supported learning of many different TNs, rather than specializing on predicting the output of single TN. After training, the weights of the LSNN remained fixed, and it was required to learn the input/output behaviour of TNs from $\mathcal{F}$ that it had never seen before in an online manner by just using its short-term memory and dynamics. See the suppl. for further details.

**Results:** Most of the functions that are computed by TNs from the class $\mathcal{F}$ are nonlinear, as illustrated in Fig. 2G for the case of inputs $(x_1, x_2)$ with $x_1 = x_2$. Hence learning the input/output behaviour of any such TN with biologically realistic local plasticity mechanisms presents a daunting challenge for a SNN. Fig. 2C shows that after a few thousand training iterations in the outer loop, the LSNN achieves low MSE for learning new TNs from the family $\mathcal{F}$, significantly surpassing the performance of an optimal linear approximator (linear regression) that was trained on all 500 pairs of inputs and target outputs, see orange curve in Fig. 2C,E. In view of the fact that each TN is defined by 40 parameters, it comes at some surprise that the resulting network learning algorithm of the LSNN for learning the input/output behaviour of a new TN produces in general a good approximation of the TN after just 5 to 20 trials, where in each trial one randomly drawn labelled example is presented. One sample of a generic learning process is shown in Fig. 2D. Each sequence of examples evokes an internal model that is stored in the short-term memory of the LSNN. Fig. 2H shows the fast evolution of internal models of the LSNN for the TN during the first trials (visualized for a 1D subset of the 2D input space). We make the current internal model of the LSNN visible by probing its prediction $C(x_1, x_2)$ for hypothetical new inputs for evenly spaced points $(x_1, x_2)$ in the domain (without allowing it to modify its short-term memory; all other inputs advance the network state according to the dynamics of the LSNN). One sees that the internal model of the LSNN is from the beginning a smooth function, of the same type as the ones defined by the TNs in $\mathcal{F}$. Within a few trials this smooth function approximated the TN quite well. Hence the LSNN had acquired during the training in the outer loop of L2L a prior for the types of functions that are to be learnt, that was encoded in its synaptic weights. This prior was in fact quite efficient, since Fig. 2E and F show that the LSNN was able to learn a TN with substantially fewer trials than a generic learning algorithm for learning the TN directly in an artificial neural network as in Fig. 2A: BP with a prior that favored small weights and biases (see end of Sec. 3 in suppl.). These results suggest that L2L is able to install some form of prior knowledge about the task in the LSNN. We conjectured that the LSNN fits internal models for smooth functions to the examples it received.

We tested this conjecture in a second, much simpler, L2L scenario. Here the family $\mathcal{F}$ consisted of all sinus functions with arbitrary phase and amplitudes between 0.1 and 5. Fig. 2I shows that the LSNN also acquired an internal model for sinus functions (made visible analogously as in Fig. 2H) in this setup from training in the outer loop. Even when we selected examples in an adversarial manner, which happened to be in a straight line, this did not disturb the prior knowledge of the LSNN.

Altogether the network learning that was induced through L2L in the LSNNs is of particular interest from the perspective of the design of learning algorithms, since we are not aware of previously documented methods for installing structural priors for online learning of a recurrent network of spiking neurons.

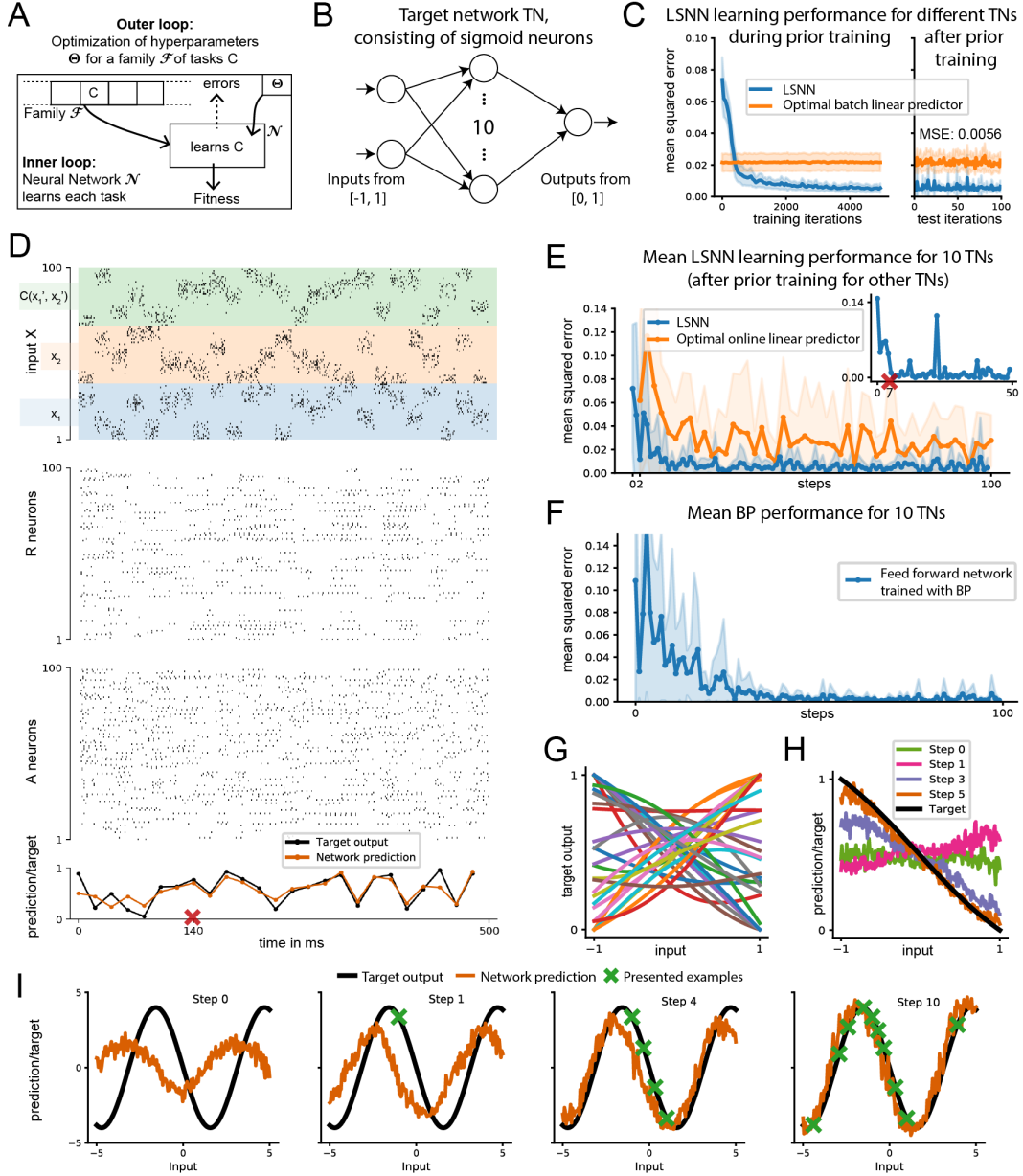

Figure 2: **LSNNs learn to learn from a teacher. A** L2L scheme for an SNN $\mathcal{N}$. **B** Architecture of the two-layer feed-forward target networks (TNs) used to generate nonlinear functions for the LSNN to learn; weights and biases were randomly drawn from [-1,1]. **C** Performance of the LSNN in learning a new TN during (left) and after (right) training in the outer loop of L2L. Performance is compared to that of an optimal linear predictor fitted to the batch of all 500 experiments for a TN. **D** Network input (top row, only 100 of 300 neurons shown), internal spike-based processing with low firing rates in the populations R and A (middle rows), and network output (bottom row) for 25 trials of 20 ms each. **E** Learning performance of the LSNN for 10 new TNs. Performance for a single TN is shown as insert, a red cross marks step 7 after which output predictions became very good for this TN. The spike raster for this learning process is the one depicted in C. Performance is compared to that of an optimal linear predictor, which, for each example, is fitted to the batch of all preceding examples. **F** Learning performance of BP for the same 10 TNs as in D, working directly on the ANN from A, with a prior for small weights. **G** Sample input/output curves of TNs on a 1D subset of the 2D input space, for different weight and bias values. **H** These curves are all fairly smooth, like the internal models produced by the LSNN while learning a particular TN. **I** Illustration of the prior knowledge acquired by the LSNN through L2L for another family $\mathcal{F}$ (sinus functions). Even adversarially chosen examples (Step 4) do not induce the LSNN to forget its prior.

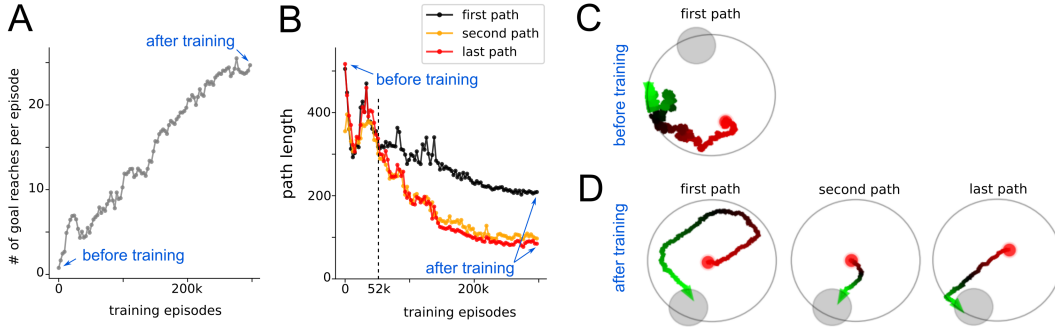

Figure 3: **Meta-RL results for an LSNN. A, B** Performance improvement during training in the outer loop. **C, D** Samples of navigation paths produced by the LSNN before and after this training. Before training, the agent performs a random walk (**C**). In this example it does not find the goal within the limited episode duration. After training (**D**), the LSNN had acquired an efficient exploration strategy that uses two pieces of abstract knowledge: that the goal always lies on the border, and that the goal position is the same throughout an episode. Note that all synaptic weights of the LSNNs remained fixed after training.

## 6 LSNNs learn-to-learn from reward

We now turn to an application of meta reinforcement learning (meta-RL) to LSNNs. In meta-RL, the LSNN receives rewards instead of teacher inputs. Meta-RL has led to a number of remarkable results for LSTM networks, see e.g. [6, 7]. In addition, [8] demonstrates that meta-RL provides a very interesting perspective of reward-based learning in the brain. We focused on one of the more challenging demos of [6] and [7], where an agent had to learn to find a target in a 2D arena, and to navigate subsequently to this target from random positions in the arena. This task is related to the well-known biological learning paradigm of the Morris water maze task [30, 31]. We study here the capability of an agent to discover two pieces of abstract knowledge from the concrete setup of the task: the distribution of goal positions, and the fact that the goal position is constant within each episode. We asked whether the agent would be able to exploit the pieces of abstract knowledge from learning for many concrete episodes, and use it to navigate more efficiently.

**Task:** An LSNN-based agent was trained on a family of navigation tasks with continuous state and action spaces in a circular arena. The task is structured as a sequence of episodes, each lasting 2 seconds. The goal was placed randomly for each episode on the border of the arena. When the agent reached the goal, it received a reward of 1, and was placed back randomly in the arena. When the agent hit a wall, it received a negative reward of -0.02 and the velocity vector was truncated to remain inside the arena. The objective was to maximize the number of goals reached within the episode. This family $\mathcal{F}$ of tasks is defined by the infinite set of possible goal positions. For each episode, an optimal agent is expected to explore until it finds the goal position, memorize it and exploits this knowledge until the end of the episode by taking the shortest path to the goal. We trained an LSNN so that the network could control the agent's behaviour in all tasks, without changing its network weights.

**Implementation:** Since LSNNs with just a few hundred neurons are not able to process visual input, we provided the current position of the agent within the arena through a place-cell like Gaussian population rate encoding of the current position. The lack of visual input made it already challenging to move along a smooth path, or to stay within a safe distance from the wall. The agent received information about positive and negative rewards in the form of spikes from external neurons. For training in the outer loop, we used BPTT together with DEEP R applied to the surrogate objective of the Proximal Policy Optimization (PPO) algorithm [32]. In this task the LSNN had 400 recurrent units (200 excitatory, 80 inhibitory and 120 adaptive neurons with adaptation time constant $\tau_a$ of 1200 ms), the network was rewired with a fixed connectivity of 20%. The resulting network diagram and spike raster is shown in Suppl. Fig. 1.

**Results:** The network behaviour before, during, and after L2L optimization is shown in Fig. 3. Fig. 3A shows that a large number of training episodes finally provides significant improvements. With a close look at Fig. 3B, one sees that before 52k training episodes, the intermediate path plan-

ning strategies did not seem to use the discovered goal position to make subsequent paths shorter. Hence the agents had not yet discovered that the goal position does not change during an episode. After training for 300k episodes, one sees from the sample paths in Fig. 3D that both pieces of abstract knowledge had been discovered by the agent. The first path in Fig. 3D shows that the agent exploits that the goal is located on the border of the maze. The second and last paths show that the agent knows that the position is fixed throughout an episode. Altogether this demo shows that meta-RL can be applied to RSNNs, and produces previously not seen capabilities of sparsely firing RSNNs to extract abstract knowledge from experimentation, and to use it in clever ways for controlling behaviour.

# 7  Discussion

We have demonstrated that deep learning provides a useful new tool for the investigation of networks of spiking neurons: It allows us to create architectures and learning algorithms for RSNNs with enhanced computing and learning capabilities. In order to demonstrate this, we adapted BPTT so that it works efficiently for RSNNs, and can be combined with a biologically inspired synaptic rewiring method (DEEP R). We have shown in section 4 that this method allows us to create sparsely connected RSNNs that approach the performance of LSTM networks on common benchmark tasks for the classification of spatio-temporal patterns (sequential MNIST and TIMIT). This qualitative jump in the computational power of RSNNs was supported by the introduction of adapting neurons into the model. Adapting neurons introduce a spread of longer time constants into RSNNs, as they do in the neocortex according to [33]. We refer to the resulting variation of the RSNN model as LSNNs, because of the resulting longer short-term memory capability. This form of short-term memory is of particular interest from the perspective of energy efficiency of SNNs, because it stores and transmits stored information through non-firing of neurons: A neuron that holds information in its increased firing threshold tends to fire less often.

We have shown in Fig. 2 that an application of deep learning (BPTT and DEEP R) in the outer loop of L2L provides a new paradigm for learning of nonlinear input/output mappings by a RSNN. This learning task was thought to require an implementation of BP in the RSNN. We have shown that it requires no BP, not even changes of synaptic weights. Furthermore we have shown that this new form of network learning enables RSNNs, after suitable training with similar learning tasks in the outer loop of L2L, to learn a new task from the same class substantially faster. The reason is that the prior deep learning has installed abstract knowledge (priors) about common properties of these learning tasks in the RSNN. To the best of our knowledge, transfer learning capabilities and the use of prior knowledge (see Fig. 2I) have previously not been demonstrated for SNNs. Fig 3 shows that L2L also embraces the capability of RSNNs to learn from rewards (meta-RL). For example, it enables a RSNN – without any additional outer control or clock – to embody an agent that first searches an arena for a goal, and subsequently exploits the learnt knowledge in order to navigate fast from random initial positions to this goal. Here, for the sake of simplicity, we considered only the more common case when all synaptic weights are determined by the outer loop of L2L. But similar results arise when only some of the synaptic weights are learnt in the outer loop, while other synapses employ local synaptic plasticity rules to learn the current task [27].

Altogether we expect that the new methods and ideas that we have introduced will advance our understanding and reverse engineering of RSNNs in the brain. For example, the RSNNs that emerged in Fig. 1-3 all compute and learn with a brain-like sparse firing activity, quite different from a SNN that operates with rate-codes. In addition, these RSNNs present new functional uses of short-term memory that go far beyond remembering a preceding input as in [34], and suggest new forms of activity-silent memory [35].

Apart from these implications for computational neuroscience, our finding that RSNNs can acquire powerful computing and learning capabilities with very energy-efficient sparse firing activity provides new application paradigms for spike-based computing hardware through non-firing.

**Acknowledgments**

This research/project was supported by the HBP Joint Platform, funded from the European Union's Horizon 2020 Framework Programme for Research and Innovation under the Specific Grant Agreement No. 720270 (Human Brain Project SGA1) and under the Specific Grant Agreement No.

785907 (Human Brain Project SGA2). We gratefully acknowledge the support of NVIDIA Corporation with the donation of the Quadro P6000 GPU used for this research. Research leading to these results has in parts been carried out on the Human Brain Project PCP Pilot Systems at the Jülich Supercomputing Centre, which received co-funding from the European Union (Grant Agreement No. 604102). We gratefully acknowledge Sandra Diaz, Alexander Peyser and Wouter Klijn from the Simulation Laboratory Neuroscience of the Jülich Supercomputing Centre for their support. The computational results presented have been achieved in part using the Vienna Scientific Cluster (VSC).

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
