[Supplementary Material]

# Supplementary information for: *Long short-term memory and learning-to-learn in networks of spiking neurons*

**Guillaume Bellec\*, Darjan Salaj\*, Anand Subramoney\*, Robert Legenstein & Wolfgang Maass**
Institute for Theoretical Computer Science
Graz University of Technology, Austria
{bellec,salaj,subramoney,legenstein,maass}@igi.tugraz.at

We provide in this supplement detailed information on the models and simulations of the main text, structured according to the corresponding sections therein.

## 2 LSNN model

**Neuron model:** In continuous time the spike trains $x_i(t)$ and $z_j(t)$ are formalized as sums of Dirac pulses. Neurons are modeled according to a standard adaptive leaky integrate-and-fire model. A neuron $j$ spikes as soon at its membrane potential $V_j(t)$ is above its threshold $B_j(t)$. At each spike time $t$, the membrane potential $V_j(t)$ is reset by subtracting the current threshold value $B_j(t)$ and the neuron enters a strict refractory period where it cannot spike again. Importantly at each spike the threshold $B_j(t)$ of an adaptive neuron is increased by a constant $\beta/\tau_{a,j}$. Then the threshold decays back to a baseline value $b_j^0$. Between spikes the membrane voltage $V_j(t)$ and the threshold $B_j(t)$ are following the dynamics

$$\tau_m \dot{V}_j(t) = -V_j(t) + R_m I_j(t) \tag{1}$$
$$\tau_{a,j} \dot{B}_j(t) = b_j^0 - B_j(t), \tag{2}$$

where $\tau_m$ is the membrane time constant, $\tau_{a,j}$ is the adaptation time constant and $R_m$ is the membrane resistance. The input current $I_j(t)$ is defined as the weighted sum of spikes from external inputs and other neurons in the network:

$$I_j(t) = \sum_i W_{ji}^{in} x_i(t - d_{ji}^{in}) + \sum_i W_{ji}^{rec} z_i(t - d_{ji}^{rec}), \tag{3}$$

where $W_{ji}^{in}$ and $W_{ji}^{rec}$ denote respectively the input and the recurrent synaptic weights and $d_{ji}^{in}$ and $d_{ji}^{rec}$ the corresponding synaptic delays. All network neurons are connected to a population of readout neurons with weights $W_{kj}^{out}$. When network neuron $j$ spikes, the output synaptic strength $W_{kj}^{out}$ is added to the membrane voltage $y_k(t)$ of all readout neurons $k$. $y_k(t)$ also follows the dynamics of a leaky integrator $\tau_m \dot{y}_k(t) = -y_k(t)$.

**Implementation in discrete time:** Our simulations were performed in discrete time with a time step $\delta t = 1$ ms. In discrete time, the spike trains are modeled as binary sequences $x_i(t), z_j(t) \in \{0, \frac{1}{\delta t}\}$, so that they converge to sums of Dirac pulses in the limit of small time steps. Neuron $j$ emits a spike at time $t$ if it is currently not in a refractory period, and its membrane potential $V_j(t)$ is above its threshold $B_j(t)$. During the refractory period following a spike, $z_j(t)$ is fixed to 0. The dynamics of the threshold is defined by $B_j(t) = b_j^0 + \beta b_j(t)$ where $\beta$ is a constant which scales the deviation $b_j(t)$ from the baseline $b_j^0$. The neural dynamics in discrete time reads as follows

$$V_j(t + \delta t) = \alpha V_j(t) + (1 - \alpha) R_m I_j(t) - B_j(t) z_j(t) \delta t \tag{4}$$
$$b_j(t + \delta t) = \rho_j b_j(t) + (1 - \rho_j) z_j(t), \tag{5}$$

where $\alpha = \exp(-\frac{\delta t}{\tau_m})$ and $\rho_j = \exp(-\frac{\delta t}{\tau_{a,j}})$. The term $B_j(t)z_j(t)\delta t$ implements the reset of the membrane voltage after each spike. The current $I_j(t)$ is the weighted sum of the incoming spikes. The definition of the input current in equation (3) holds also for discrete time, with the difference that spike trains now assume values in $\{0, \frac{1}{\delta t}\}$.

## 3 Applying BPTT with DEEP R to RSNNs and LSNNs

**Propagation of gradients in recurrent networks of LIF neurons:** In artificial recurrent neural networks such as LSTMs, gradients can be computed with backpropagation through time (BPTT). For BPTT in spiking neural networks, complications arise from the non-differentiability of the output of spiking neurons, and from the fact that gradients need to be propagated either through continuous time or through many time steps if time is discretized. Therefore, in [1, 2] it was proposed to use a pseudo-derivative.

$$\frac{dz_j(t)}{dv_j(t)} := \max\{0, 1 - |v_j(t)|\}, \tag{6}$$

where $v_j(t)$ denotes the normalized membrane potential $v_j(t) = \frac{V_j(t) - B_j(t)}{B_j(t)}$. This made it possible to train deep feed-forward networks of deterministic binary neurons [1, 2]. We observed that this convention tends to be unstable for very deep (unrolled) recurrent networks of spiking neurons. To achieve stable performance we dampened the increase of back propagated errors through spikes by using a pseudo-derivative of amplitude $\gamma < 1$ (typically $\gamma = 0.3$):

$$\frac{dz_j(t)}{dv_j(t)} := \gamma \max\{0, 1 - |v_j(t)|\}. \tag{7}$$

Note that in adaptive neurons, gradients can propagate through many time steps in the dynamic threshold. This propagation is not affected by the dampening.

**Rewiring and weight initialization of excitatory and inhibitory neurons:** In all experiments except those reported in Fig. 2, the neurons were either excitatory or inhibitory. When the neuron sign were not constrained, the initial network weights were drawn from a Gaussian distribution $W_{ji} \sim \frac{w_0}{\sqrt{n_{in}}} \mathcal{N}(0, 1)$, where $n_{in}$ is the number of afferent neurons in the considered weight matrix (i.e., the number of columns of the matrix), $\mathcal{N}(0, 1)$ is the zero-mean unit-variance Gaussian distribution and $w_0$ is a weightscaling factor chosen to be $w_0 = \frac{1\text{Volt}}{R_m}\delta t$. With this choice of $w_0$ the resistance $R_m$ becomes obsolete but the vanishing-exploding gradient theory [3, 4] can be used to avoid tuning by hand the scaling of $W_{ji}$. In particular the scaling $\frac{1}{\sqrt{n_{in}}}$ used above was sufficient to initialize networks with realistic firing rates and that can be trained efficiently.

When the neuron sign were constrained, all outgoing weights $W_{ji}^{rec}$ or $W_{ji}^{out}$ of a neuron $i$ had the same sign. In those cases, DEEP R [5] was used as it maintains the sign of each synapse during training. The sign is thus inherited from the initialization of the network weights. This raises the need of an efficient initialization of weight matrices for given fractions of inhibitory and excitatory neurons. To do so, a sign $\kappa_i \in \{-1, 1\}$ is generated randomly for each neuron $i$ by sampling from a Bernoulli distribution. The weight matrix entries are then sampled from $W_{ji} \sim \kappa_i |\mathcal{N}(0, 1)|$ and post-processed to avoid exploding gradients. Firstly, a constant is added to each weight so that the sum of excitatory and inhibitory weights onto each neuron $j$ ($\sum_i W_{ji}$) is zero [6] (if $j$ has no inhibitory or no excitatory incoming connections this step is omitted). To avoid exploding gradients it is important to scale the weight so that the largest eigenvalue is lower of equal to 1 [3]. Thus, we divided $W_{ji}$ by the absolute value of its largest eigenvalue. When the matrix is not square, eigenvalues are ill-defined. Therefore, we first generated a large enough square matrix and selected the required number of rows or columns with uniform probabilities. The final weight matrix is scaled by $w_0$ for the same reasons as before.

To initialize matrices with a sparse connectivity, dense matrices were generated as described above and multiplied with a binary mask. The binary mask was generated by sampling uniformly the neuron coordinates that were non-zero at initialization. DEEP R maintains the initial connectivity level throughout training by dynamically disconnecting synapses and reconnecting others elsewhere. The $L_1$-norm regularization parameter of DEEP R was set to $0.01$ and the temperature parameter of DEEP R was left at $0$.

# 4   Computational performance of LSNNs

**MNIST setup:**   The pixels of an MNIST image were presented sequentially to the LSNN in $784$ time steps. Two input encodings were considered. First, we used a population coding where the grey scale value (which is in the range $[0, 1]$) of the currently presented pixel was directly used as the firing probability of each of the $80$ input neurons in that time step.

In a second type of input encoding – that is closer to the way how spiking vision sensors encode their input – each of the $80$ input neurons was associated with a particular threshold for the grey value, and this input neuron fired whenever the grey value of the currently presented pixel crossed its threshold. Here, we used two input neurons per threshold, one spiked at threshold crossings from below, and one at the crossings from above. This input convention was chosen for the LSNN results of Fig. 1.B.

The output of the network was determined by averaging the readout output over the $56$ time steps following the presentation of the digit. The network was trained by minimizing the cross entropy error between the softmax of the averaged readout and the label distributions. The best performing models use rewiring with a global connectivity level of $12\%$ was used during training to optimize a sparse network connectivity structure (i.e., when randomly picking two neurons in the network, the probability that they would be connected is $0.12$). This implies that only a fraction of the parameters were finally used as compared to a similarly performing LSTM network.

Tables S1 and S2 contain the results and details of training runs where each time step lasted for $1$ ms and $2$ ms respectively.

| Model | # neurons | conn. | # params | # runs | mean | std. | max. |
|---|---|---|---|---|---|---|---|
| LSTM | 128 | 100% | 67850 | 12 | 79.8% | 26.6% | 98.5% |
| RNN | 128 | 100% | 17930 | 10 | 71.3% | 24.5% | 89% |
| LSNN | 100(A), 120(R) | 12% | 8185 (full 68210) | 12 | 94.2% | 0.3% | 94.7% |
| LSNN | 100(A), 200(R) | 12% | 14041 (full 117010) | 1 | - | - | 95.7% |
| LSNN | 350(A), 350(R) | 12% | 66360 (full 553000) | 1 | - | - | 96.1% |
| LSNN | 100(A), 120(R) | 100% | 68210 | 10 | 92.0% | 0.7% | 93.3% |
| LIF | 220 | 100% | 68210 | 10 | 60.9% | 2.7% | 63.3% |

Table S1: Results on the sequential MNIST task when each pixel is displayed for 1ms. For an LSN, DEEP R is used to optimize the network under a sparse connectivity constraint, we report the number of parameters including and not including the disconnected synapses.

| Model | # neurons | conn. | # params | # runs | mean | std. | max. |
|---|---|---|---|---|---|---|---|
| LSTM | 128 | 100% | 67850 | 12 | 48.2% | 39.9% | 98.0% |
| RNN | 128 | 100% | 17930 | 12 | 30% | 23.6% | 67.9% |
| LSNN | 100(A), 120(R) | 12% | 8185 (full 68210) | 12 | 93.8% | 5.8% | 96.4% |
| LSNN | 350(A), 350(R) | 12% | 66360 (full 553000) | 1 | - | - | 97.1% |
| LSNN | 100(A), 120(R) | 100% | 68210 | 10 | 90.5% | 1.4% | 93.7% |
| LIF | 220 | 100% | 68210 | 11 | 34.6% | 8.8% | 51.8% |

Table S2: Results on the sequential MNIST task when each pixel is displayed for 2ms.

**TIMIT setup:**   To investigate if the performance of LSNNs can scale to real world problems, we considered the TIMIT speech recognition task. We focused on the frame-wise classification where the LSNN has to classify each audio-frame to one of the 61 phoneme classes.

We followed the convention of Halberstadt [7] for grouping of training, validation, and testing sets ($3696$, $400$, and $192$ sequences respectively). The performance was evaluated on the *core test set* for consistency with the literature. Raw audio is preprocessed into 13 Mel Frequency Cepstral Coefficients (MFCCs) with frame size 10 ms and on input window of 25 ms. We computed the first and the second order derivatives of MFCCs and combined them, resulting in 39 input channels. These 39 input channels were mapped to 39 input neurons which unlike in MNIST emit continuous values $x_i(t)$ instead of spikes, and these values were directly used in equation 3 for the currents of the postsynaptic neurons.

Since we simulated the LSNN network in 1 ms time steps, every input frame which represents 10 ms of the input audio signal was fed to the LSNN network for 10 consecutive 1 ms steps. The softmax output of the LSNN was averaged over every 10 steps to produce the prediction of the phone in the current input frame. The LSNN was rewired with global connectivity level of 20%.

**Parameter values:** For adaptive neurons, we used $\beta_j = 1.8$, and for regular spiking neurons we used $\beta_j = 0$ (i.e. $B_j$ is constant). The baseline threshold voltage was $b_j^0 = 0.01$ and the membrane time constant $\tau_m = 20$ ms. Networks were trained using the default Adam optimizer, and a learning rate initialized at $0.01$. The dampening factor for training was $\gamma = 0.3$.

For sequential MNIST, all networks were trained for 36000 iterations with a batch size of 256. Learning rate was decayed by a factor 0.8 every 2500 iterations. The adaptive neurons in the LSNN had an adaptation time constant $\tau_a = 700$ ms (1400 ms) for 1 ms (2 ms) per pixel setup. The baseline artificial RNN contained 128 hidden units with the hyperbolic tangent activation function. The LIF network was formed by a fully connected population of 220 regular spiking neurons.

For TIMIT, the LSNN network consisted of 300 regular neurons and 100 adaptive neurons which resulted in approximately 400000 parameters. Network was trained for 80 epochs with batches of 32 sequences. Adaptation time constant of adaptive neurons was set to $\tau_a = 200$ ms. Refractory period of the neurons was set to 2 ms, the membrane time constant of the output Y neurons to 3 ms, and the synaptic delay was randomly picked from $\{1, 2\}$ ms.

We note that due to the rewiring of the LSNN using DEEP R [5] method, only a small fraction of the weights had non-zero values (8185 in MNIST, $\sim 80000$ in TIMIT).

## 5   LSNNs learn-to-learn from a teacher

**Experimental setup:**

*Function families:* The LSNN was trained to implement a regression algorithm on a family of functions $\mathcal{F}$. Two specific families were considered: In the first function family, the functions were defined by feed-forward neural networks with 2 inputs, 1 hidden layer consisting of 10 hidden neurons, and 1 output, where all the parameters (weights and biases) were chosen uniformly randomly between $[-1, 1]$. The inputs were between $[-1, 1]$ and the outputs were scaled to be between $[0, 1]$. We call these networks Target Networks (TNs). In the second function family, the targets were defined by sinusoidal functions $y = A \sin(\phi + x)$ over the domain $x \in [-5, 5]$. The specific function to be learned was defined then by the phase $\phi$ and the amplitude $A$, which were chosen uniformly random between $[0, \pi]$ and $[0.1, 5]$ respectively.

*Input encoding:* Analog values were transformed into spiking trains to serve as inputs to the LSNN as follows: For each input component, 100 input neurons are assigned values $m_1, \ldots m_{100}$ evenly distributed between the minimum and maximum possible value of the input. Each input neuron has a Gaussian response field with a particular mean and standard deviation, where the means are uniformly distributed between the minimum and maximum values to be encoded, and with a constant standard deviation. More precisely, the firing rate $r_i$ (in Hz) of each input neuron $i$ is given by $r_i = r_{max} \exp\left(-\frac{(m_i - z_i)^2}{2\sigma^2}\right)$, where $r_{max} = 200$ Hz, $m_i$ is the value assigned to that neuron, $z_i$ is the analog value to be encoded, and $\sigma = \frac{(m_{max} - m_{min})}{1000}$, $m_{min}$ with $m_{max}$ being the minimum and maximum values to be encoded.

*LSNN setup and training schedule:* The standard LSNN model was used, with 300 hidden neurons for the TN family of learning tasks, and 100 for the sinusoidal family. Of these, $40\%$ were adaptive in all simulations. We used all-to-all connectivity between all neurons (regular and adaptive). The output of the LSNN was a linear readout that received as input the mean firing rate of each of the neurons per step i.e the number of spikes divided by 20 for the 20 ms time window that the step consists of.

The network training proceeded as follows: A new target function was randomly chosen for each **episode** of training, i.e., the parameters of the target function are chosen uniformly randomly from within the ranges above (depending on whether its a TN or sinusoidal). Each **episode** consisted of a sequence of 500 **steps**, each lasting for 20 ms. In each step, one training example from the current

function to be learned was presented to the LSNN. In such a step, the inputs to the LSNN consisted of a randomly chosen vector $\mathbf{x}$ with its dimensionality $d$ and range determined by the target function being used ($d = 2$ for TNs, $d = 1$ for sinusoidal target function). In addition, at each step, the LSNN also got the target value $C(\mathbf{x}')$ from the previous step, i.e., the value of the target calculated using the target function for the inputs given at the previous step (in the first step, $C(\mathbf{x}')$ is set to $\mathbf{0}$).

All the weights of the LSNN were updated using our variant of BPTT, once per **iteration**, where an **iteration** consists of a batch of 10 **episodes**, and the weight updates are accumulated across episodes in an iteration. The Adam [8] variant of BP was used with standard parameters and a learning rate of 0.001. The loss function for training was the mean squared error (MSE) of the LSNN predictions over an iteration (i.e. over all the steps in an episode, and over the entire batch of episodes in an iteration). In addition, a regularization term was used to maintain a firing rate of 20 Hz. Specifically, the regularization term $R$ is defined as the mean squared difference between the average neuron firing rate in the LSNN and a target of 20 Hz. The total loss $L$ was then given by $L = MSE + 30\,R$. In this way, we induce the LSNN to use sparse firing. We trained the LSNN for 5000 iterations in all cases.

**Parameter values:**   The LSNN parameters were as follows: 5 ms neuronal refractory period, delays spread uniformly between $0 - 5$ ms, adaptation time constants of the adaptive neurons spread uniformly between $1 - 1000$ ms, $\beta = 1.6$ for adaptive neurons (0 for regular neurons), membrane time constant $\tau = 20$ ms, 0.03 mV baseline threshold voltage. The dampening factor for training was $\gamma = 0.4$.

**Analysis and comparison:**   The linear baseline was calculated using linear regression with L2 regularization with a regularization factor of 100 (determined using grid search), using the mean spiking trace of all the neurons. The mean spiking trace was calculated as follows: First the neuron traces were calculated using an exponential kernel with 20 ms width and a time constant of 20 ms. Then, for every step, the mean value of this trace was calculated to obtain the mean spiking trace. In Fig. 2C, for each episode consisting of 500 steps, the mean spiking trace from a random subset of 450 steps was used to train the linear regressor, and the mean spiking trace from remaining 50 steps was used to calculate the test error. The reported baseline is the mean of the test error over one batch of 10 episodes with error bars of one standard deviation. In Fig. 2E, for each episode, after every step $k$, the mean spiking traces from the first $k - 1$ steps were used to train the linear regressor, and the test error was calculated using the mean spiking trace for the $k$th step. The reported baseline is a mean of the test error over one batch of 10 episodes with error bars of one standard deviation.

For the case where neural networks defined the function family, the total test MSE was $0.0056 \pm 0.0039$ (linear baseline MSE was $0.0217 \pm 0.0046$). For the sinusoidal function family, the total test MSE was $0.3134 \pm 0.2293$ (linear baseline MSE was $1.4592 \pm 1.2958$).

*Comparison with backprop:* The comparison was done for the case where the LSNN is trained on the function family defined by target networks. A feed-forward (FF) network with 10 hidden neurons and 1 output was constructed. The input to this FF network were the analog values that were used to generate the spiking input and targets for the LSNN. Therefore the FF had 2 inputs, one for each of $x_1$ and $x_2$. The error reported in Fig 2F is the mean training error over 10 batches with error bars of one standard deviation.

The FF network was initialized with Xavier normal initialization [9] (which had the best performance, compared to Xavier uniform and plain uniform between $[-1, 1]$). Adam [8] with AMSGrad [10] was used with parameters $\eta = 10^{-1}, \beta_1 = 0.7, \beta_2 = 0.9, C = 10^{-5}$. These were the optimal parameters as determined by a grid search. Together with the Xavier normal initialization and the weight regularization parameter $C$, the training of the FF favoured small weights and biases.

# 6   LSNNs learn-to-learn from reward

**Experimental setup:**

*Task family:* An LSNN-based agent was trained on a family of navigation tasks in a two dimensional circular arena. For all tasks, the arena is a circle with radius 1 and goals are smaller circles of radius 0.3 with centres uniformly distributed on the circle of radius 0.85. At the beginning of an episode

Supplementary Figure S1: **Meta-RL results for an LSNN. A** Samples of paths after training. **B** Connectivity between sub-populations of the network after training. The global connectivity in the network was constrained to $20\%$. **C** The network dynamics that produced the behavior shown in A. Raster plots and thresholds are displayed as in Fig. 1.D, only 1 second and 100 neurons are shown in each raster plots.

and after the agent reaches a goal, the agent's position is set randomly with uniform probability within the arena. At every timestep, the agent chooses an action by generating a small velocity vector of Euclidean norm smaller or equal to $a_{scale} = 0.02$. When the agent reaches the goal, it receives a reward of $1$. If the agent attempts to move outside the arena, the new position is given by the intersection of the velocity vector with the border and the agent receives a negative reward of $-0.02$.

*Input encoding:* Information of the current environmental state $s(t)$ and the reward $r(t)$ were provided to the LSNN at each time step $t$ as follows: The state $s(t)$ is given by the $x$ and $y$ coordinate of the agent's position (see top of Fig. S1C). Each position coordinate $\xi(t) \in [-1, 1]$ is encoded by $40$ neurons which spike according to a Gaussian population rate code defined as follows: a preferred coordinate value $\xi_i$, is assigned to each of the $40$ neurons, where $\xi_i$'s are evenly spaced between $-1$ and $1$. The firing rate of neuron $i$ is then given by $r_{max} \exp(-100(\xi_i - \xi)^2)$ where $r_{max}$ is 500 Hz. The instantaneous reward $r(t)$ is encoded by two groups of $40$ neurons (see green row at the top of Fig. S1C). All neuron in the first group spike in synchrony each time a reward of $1$ is received (i.e., the goal was reached), and the second group spikes when a reward of $-0.02$ is received (i.e., the agent moved into a wall).

*Output decoding:* The output of the LSNN is provided by five readout neurons. Their membrane potentials $y_i(t)$ define the outputs of the LSNN. The action vector $\mathbf{a}(t) = (a_x(t), a_y(t))^T$ is sampled from the distribution $\pi_\theta$ which depends on the network parameters $\theta$ through the readouts $y_i(t)$ as follows: The coordinate $a_x(t)$ ($a_y(t)$) is sampled from a Gaussian distribution with mean $\mu_x = \tanh(y_1(t))$ ($\mu_y = \tanh(y_2(t))$) and variance $\phi_x = \sigma(y_3(t))$ ($\phi_y = \sigma(y_4(t))$). The velocity vector that updates the agent's position is then defined as $a_{scale}\,\mathbf{a}(t)$. If this velocity has a norm larger than $a_{scale}$, it is clipped to a norm of $a_{scale}$.

The last readout output $y_5(t)$ is used to predict the value function $V_\theta(t)$. It estimates the expected discounted sum of future rewards $R(t) = \sum_{t' > t} \eta^{t' - t} r(t')$, where $\eta = 0.99$ is the discount factor and $r(t')$ denotes the reward at time $t'$. To enable the network to learn complex forms of exploration we introduced current noise in the neuron model in this task. At each time step, we added a small

Gaussian noise with mean 0 and standard deviation $\frac{1}{R_m}\nu_j$ to the current $I_j$ into neuron $j$. Here, $\nu_j$ is a network parameter initialized at 0.03 and optimized by BPTT alongside the network weights.

**Network training:** To train the network we used the Proximal Policy Optimization algorithm (PPO) [11]. For each training iteration, $K$ full episodes of $T$ timesteps were generated with fixed parameters $\theta_{old}$ (here $K = 10$ and $T = 2000$). We write the clipped surrogate objective of PPO as $O^{PPO}(\theta_{old}, \theta, t, k)$ (this is defined under the notation $L^{CLIP}$ in [11]). The loss with respect to $\theta$ is then defined as follows:

$$\mathcal{L}(\theta) \quad = \quad -\frac{1}{KT}\sum_{k<K}\sum_{t<T}O^{PPO}(\theta_{old},\theta,t,k) + \mu_v\left(R(t,k) - V_\theta(t,k)\right)^2 \tag{8}$$

$$-\mu_e H(\pi_\theta(k,t)) + \mu_{firing}\frac{1}{n}\sum_j ||\frac{1}{KT}\sum_{k,t}z_j(t,k) - f^0||^2, \tag{9}$$

where $H(\pi_\theta)$ is the entropy of the distribution $\pi_\theta$, $f^0$ is a target firing rate of 10 Hz, and $\mu_v$, $\mu_e$, $\mu_{firing}$ are regularization hyper-parameters. Importantly probability distributions used in the definition of the loss $\mathcal{L}$ (i.e. the trajectories) are conditioned on the current noises, so that for the same noise and infinitely small parameter change from $\theta_{old}$ to $\theta$ the trajectories and the spike trains are the same. At each iteration this loss function $\mathcal{L}$ is then minimized with one step of the ADAM optimizer.

**Parameter values:** In this task the LSNN had 400 hidden units (200 excitatory neurons, 80 inhibitory neurons and 120 adaptive neurons with adaptation time constants $\tau_a = 1200$ ms) and the network was rewired with a fixed global connectivity of 20% [5]. The membrane time constants were similarly sampled between 15 and 30 ms. The adaptation amplitude $\beta$ was set to 1.7. The refractory period was set to 3 ms and delays were sampled uniformly between 1 and 10 ms. The regularization parameters $\mu_v$, $\mu_e$ and $\mu_{firing}$ were respectively 1, 0.001, and 100. The parameter $\epsilon$ of the PPO algorithm was set to 0.2. The learning rate was initialized to 0.01 and decayed by a factor 0.5 every 5000 iterations. We used the default parameters for ADAM, except for the parameter $\epsilon$ which we set to $10^{-5}$.