[Reviews · NeurIPS 2018]

Reviewer 1



Summary Recurrent networks of leaky integrate-and-fire neurons with (spike frequency) adaptation are trained with backpropagation-through-time (adapted to spiking neurons) to perform digit recognition (temporal MNIST), speech recognition (TIMIT), learning to learn simple regression tasks and learning to find a goal location in simple navigation tasks. The performances on temporal MNIST and TIMIT are similar to the one of LSTM-networks. The simple regression and navigation task demonstrate that connection weights exist that allow to solve simple tasks using the short-term memory of spiking neurons with adaptation, without the need of ongoing synaptic plasticity. Quality The selection of tasks is interesting, the results are convincing and the supplementary information seems to provide sufficient details to reproduce them. Clarity The paper communicates the main messages clearly. But the writing could be improved significantly. The introduction is not well structured. Why are the contributions in paragraphs 2 and 4 separated by the long paragraph on L2L? What is meant by the two levels of learning and representation (line 43)? Why is the discussion of the PFC and STDP relevant for the rest of the paper (lines 48 to 56)? I did not understand the sentence that starts on line 123. Fig 1B seems to indicate that the LSNN performs similarly as an LSTM, not significantly better. line 128: It would be helpful say here already, how the inputs are encoded for the spiking networks such that the reader does not need to go to the supplementary material to understand this aspect. line 180: I did not understand how the network could learn to cheat. Originality The work seems original. But a section on related work is missing. There are also other works that try to solve (temporal) MNIST and TIMIT with networks of spiking neurons (e.g. https://doi.org/10.1016/j.neucom.2013.06.052). How is this work related to other approaches? Significance I enjoyed reading this work. It is nice to see networks of adaptive spiking neurons perform well on temporal MNIST and TIMIT and solve the simple regression and navigation tasks without synaptic plasticity (after having learned the domain knowledge). If the approach scales, it may also be interesting for neuromorphic hardware. What we learn about biological neural networks is less clear. Activity or short-term plasticity dependent models of working memory are well known in computational neuroscience (see e.g. http://dx.doi.org/10.1016/j.conb.2013.10.008) and the present work does not seem to make clear testable predictions beyond the observation that these networks are sufficient to solve the studied tasks. Minor Points: - Why was the same adaptation time constant used for all neurons in the first two tasks, but a distributions of time constants for the second two tasks? ==== After reading the author's response and the other reviews I am more convinced that this is exciting and novel work. But its presentation and embedding in the literature can clearly be improved. Regarding biological neural networks: I agree with the authors that the work demonstrates the power of adaptation in recurrent networks of spiking neurons. But the hypothesis that adaptation could be used as a form of activity-silent working memory is very straightforward and it is not clear to me, why nature would have favored this over e.g. activity-silent short-term-plasticity based working memory [Mongillo et al. 2008].

Reviewer 2



Summary The authors trained spiking neural networks to perform sequential MNIST, TIMIT, learning to learn families of non-linear transformations, and meta-reinforcement learning of water mazes. To accomplish this, they trained a population of LIF neurons using BPTT with smooth pseudo-derivatives around the spikes and the deep rewiring algorithm for tuning network connectivity. Some of the LIF neurons had an adapting property whereby their excitability was reduced after firing. Positives The idea of doing sequence classification, L2L, and meta-rl problems in spiking networks is very appealing, because of the applications in modeling of biological neural activity and neuromorphic computing. The tasks are compelling and the results seem strong. Even though the critical learning algorithm, BPTT, doesn't have a biological interpretation, the fact that the authors are able to solve these tasks with RSNNs at all appears to be a technical feat and a useful demonstration. Areas for Improvement However, the paper doesn't provide a historical summary of the state of the art for RSNNs applied to these sorts of problems, so as someone without a working knowledge of that literature, I'm not able to evaluate how large of an advance the paper makes beyond prior work. Further, the paper doesn't clearly explain the motivation behind the architecture and learning algorithm design. What's the role of the adaptive LIF units? Do they help with long-term dependencies somehow? Why does the architecture have "Long short term memory" in its name? Is it to do with the adaptive units, rewiring algorithm, something else? What are the essential differences between this architecture and, say, a homogenous population of LIF neurons, which make it better suited to tasks with long-term dependencies? Does deep rewiring matter -- what happens without it? In order to understand the implications of this work, these issues need to be made clear. Summary I think the quality of the work behind this paper is probably strong, but the lack of clarity makes it difficult to understand the implications of the work and to assess its originality and significance. I think an improved draft on this work could be quite a strong paper, but I'll have to recommend rejection for now. Edit after author feedback and reviewer discussion: In their rebuttal the authors explained the novelty of the work and justified some design/naming choices I asked about in my review. However, the rebuttal didn't indicate exactly how they would fix the clarity problems that led me to need to ask about those two points. Overall this seems to me like valuable work written in a somewhat unsatisfactory manner. I increased my rating to 6 to reflect the apparent quality of the underlying work, and hope the authors improve the clarity for the camera ready.

Reviewer 3



This paper aims to combine deep learning techniques (backpropagation through time and learning to learning) with recurrent networks of spiking neurons, which are more biologically realistic but typically have worse performance and learning capabilities than deep neural networks. They propose a modified version of recurrent spiking neural networks (RSNNs) that includes neural adaptation and BPTT (modified to work with spiking networks), which they term long short-term memory spiking neural networks (LSNNs). They test on two standard benchmark SL tasks: sequential MNIST and TIMIT, and find performance comparable to that of an LSTM. They also test the learning-to-learn capabilities of LSNNs by looking at two task domains: 1) function learning, where the function to be learned is the class of all continuous functions of two variables that can be computed by a 2-layer ANN, as well as simple sine functions, and 2) meta-reinforcement learning in a navigation task, where the goal is placed randomly every episode. In all, I really enjoyed this paper. It addresses an important disconnect between our understanding of how the brain works, and state of the art deep neural network architectures, which presumably are implementing at least some similar cognitive functions such as visual recognition or sequence processing. The tasks they selected are appropriate, ranging from standard benchmarking tasks in SL to toy tasks which allow for more detailed analysis, and finally a larger scale RL task. I mainly have a few concerns with regard to clarity and novelty, although I suspect these could be readily addressed in revision. 1. The paper currently feels as if it’s almost 2 separate papers put together, one about LSNNs and one about learning to learn. Are these two contributions completely orthogonal, or is there some interaction? How would LSNNs perform on for example omniglot classification or another benchmark task in which meta-learning is required (perhaps the one used in Santoro et al, 2016 on MANNs)? 2. With regard to clarity, I got confused while reading section 3, because I had expected these to be trained in the “meta-learning” sense, as the previous paragraph (end of section 2) discussed learning-to-learn at length. Going back to my point 1, it’d make much more sense of these tasks were also “meta-learning” tasks. 3. What is the relative importance of the adaptive threshold vs the modified BPTT procedure (with DEEP R) to the performance of the model? 4. It’s important to spell out what are the new contributions made in this paper vs Bellec et al, 2018, which is cited for DEEP R. Without reading this other paper, I can’t tell what is the proposed contribution (was it applied to only FF networks in the other paper?) 5. I’m curious about the disadvantages/advantages of LSNNs vs traditional DNNs. It seems like LSNNs might deal well with sequential data, but what about data without sequential structure, like images? Convnets can capitalize on spatial properties like translational invariance, etc, but would LSNNs always struggle with these? LSTMs are definitely not the first model you’d reach for when classifying MNIST. Update after reading the authors' rebuttal: While I still think that the paper is pretty strong, I'm pretty disappointed in the response, which never stated what the authors would specifically revise to address the points raised, even with regard to easily addressable issues like clarity. Because of this, I have less confidence that the paper could be revised to be in the top 50% of papers accepted to NIPS. However my opinion is that it should still be accepted.